# Europeanised Attitudes, Nationalised Communication? Evidence on the Patterns behind Political Communication Output in Brussels

**Jan Georg Plavec *** and **Barbara Pfetsch ***

Fachbereich Politik- und Sozialwissenschaften, Institut für Publizistik- und Kommunikationswissenschaft, Freie Universität Berlin, 14195 Berlin, Germany
* Correspondence: plavec@zedat.fu-berlin.de (J.G.P.); barbara.pfetsch@fu-berlin.de (B.P.)

**Abstract:** Studies of a communication deficit in the European Union (EU) have hardly taken a systematic look at the site where most of the political communication output is being created: within the elite bubble of EU politicians and correspondents in Brussels. This study builds on the communication culture approach to describe and explain the basic attitudinal patterns of EU politicians and journalists who critically shape the political communication output coming out of Brussels that is being consumed by European citizens. Based on a survey with more than 300 participating politicians and journalists, this study demonstrates that the internationalised communication context in Brussels reduces differences between the attitudes of actors from professional and national groups. We demonstrate that there is a tendency toward common elitist attitudes, complemented by a highly negative view of the public and a cynical mode of political communication. However, we observe predominantly national contact networks in Brussels and partly differing attitudes among some sub-groups of politicians and journalists, reflecting the partly conflicting national configurations of the European political and media system and the principal-agent relationships of EU politicians and journalists with their constituencies and media outlets.

**Keywords:** political communication; communication culture; journalism

## 1. Introduction

Among scholars of European communication and democracy, the EU communication deficit has been understood as a failure "to communicate with the press effectively, making it more likely still that the media will fall back on national interpretations of what is happening" (Rowinski 2017, p. 49). The structural constellations leading to this deficit and its political implications have been extensively discussed some 20 years ago, leading to the insight that there is no European public sphere but segmented European public spheres (Erbe 2005, p. 76). However, this perspective can neither fully explain the output in the actual news media nor has it been helpful in finding a real solution to tackle the communication deficit itself. Why do politicians and journalists stick to the "national lens" when communicating EU politics?

Despite attempts to look for a Europeanisation of national publics, none of the identified shortcomings have really been addressed—quite surprisingly, given the rise of populist parties who apparently make use of them (Rauh et al. 2020; Aalberg and de Vreese 2017; Stanyer 2007). Scholarly attention has never systematically turned to the actors in Brussels who produce the actual political messages for mass media. We argue that a focus on the actors of political communication is key to fully understanding why transnational EU politics is being reported through a national lens and therefore understanding the communication deficit itself. Politicians and journalists in Brussels and their basic attitudes toward political EU communication are the missing link between structure and communication output as it is received by European citizens.

The political communication culture concept is suited to fill this knowledge gap. While it has been primarily been used to compare national communication cultures (Pfetsch 2014b), we utilise it for the first time in a transnational communication context and focus on the political-communicative "milieu" (Balcytiene 2007, p. 80) in Brussels that produces these messages and therefore contributes to and sustains the communication deficit. We produced evidence of these patterns by analysing data from a survey among 140 high-level politicians and 169 journalists in Brussels. The communication culture approach helps to analyse how this unique milieu creates political messages because it focuses on the attitudes guiding their interactions. With its focus on the differences and similarities of those attitudes, this framework enables us to understand how EU communication elites act and why.

This paper analyses the strategies that politicians and journalists apply when they formulate political messages and try to get them through to their audience. In the political communication culture concept, creating communication output is understood as a common task of politicians and journalists who address the same audience. We analyse the way that they frame and distribute political messages that are to be published in national and international media outlets and the strategies behind them. Communication cultures can be characterised by analysing and comparing the basic attitudinal constellations between politicians and journalists, and they differ strongly within Europe (Pfetsch 2004, 2014a).

Our analysis cannot be separated from the structural configurations within which EU politicians and journalists operate. Brussels is a very interesting context in this respect: A classical principal-agent approach would focus on the individuals' dependencies on the media organisations with their (predominantly) national markets and audiences and the politicians' orientation toward national interests, governments, and constituencies (Michailidou and Trenz 2014; Seoane Pérez 2013) to explain the national orientation in the communicative behaviour of politicians and journalists. The communication culture approach adds to this the significance of the communication system or environment, which in this case is the European quarter in Brussels and its milieu of EU politicians and journalists.

Brussels is a highly internationalised place to work, bringing together professionals from very diverse national backgrounds and communication cultures which forces them to cooperate on common communicative and cultural ground. It is unclear how politicians and journalists fulfil their rather "national" communicative tasks and to what extent they stick to their specific communication cultures and routines while, at the same time, they have to socialise in a multinational and cosmopolitan setting that incentivises cooperation and the ability to adapt a "Europeanised" understanding of EU politics (Melchior 2017, p. 42; Chatzistavrou 2013; Mancini et al. 2007, p. 127f).

Our first central question, therefore, is whether politicians and journalists in Brussels overcome their conflicting national communication goals and styles to create a common understanding of political EU communication. Can EU politicians and journalists in Brussels create what Michailidou and Trenz (2021, p. 3) call "collaborative schemes and shared interpretative frames that bridge national media systems and languages"? The second important aspect within the communication culture concept and in this paper is the differing approaches to political communication of the professional groups themselves: Does the Brussels communication environment reduce or increase the conflicts between politicians and journalists? Integrating these two dimensions helps to understand the attitudinal constellations and the mechanisms behind the political communication output coming from Brussels.

The paper is structured as follows: First, we describe the concept of political communication culture and possible configurations of its output dimension based on a systematic literature review. After defining the relevant items of the output dimension, we analyse the respective attitudes of politicians and journalists in Brussels. We will close with a discussion of our findings, focusing primarily on the potential for a further Europeanisation of EU political communication output.

## 2. Literature Review

Political EU communication is characterised by national pillarization ("nationale Versäulung", Tobler 2002, p. 269). From a pro-integration perspective, the deficit encompasses, more generally, a dysfunctional outcome of political messages created in Brussels and disseminated via the media to a fragmented audience, ultimately resulting in a democracy deficit (Heinderyckx 2015; Spanier 2012; Hepp et al. 2009, pp. 49–52; Ward 2004).

Studies on the political EU communication have often dealt with the shortcomings of a European public sphere or the Europeanisation of national public spheres (Shipkova 2017; Pfetsch and Heft 2015; Seoane Pérez 2013; Brüggemann et al. 2009; Brüggemann and Schulz-Forberg 2009; Neidhardt 2006; Eriksen 2005; Gerhards 2000). These approaches compare systematic constellations such as the political, economic, or regulatory approaches of political communication in Europe. They look for tendencies leading (or not) to an integration of these different constellations into a common European communication system or public sphere and have in some way proven to be a dead end, partly due to a multi-segmentation of European public spheres (Erbe 2005; Koopmans and Erbe 2004).

A more fruitful approach looks for evidence inside the European quarter, where political EU communication regularly takes shape. Despite the digital transformation with its new channels of communication and the renewed uncertainty of what journalism even is (Deuze and Witschge 2018), EU politics is still predominantly being reported and discussed in the traditional print and audio-visual media or their respective online platforms as the (although beleaguered) "gatekeeper of democracy" (Rowinski 2021, p. 4). European audiences continue to perceive traditional media outlets as particularly trustworthy (European Parliament, Directorate-General for Communication 2022, p. 37). These mechanisms, as well as the "microcosm" where politicians and journalists meet each other on a regular basis (Leppik et al. 2007, p. 58; Baisnée 2007, p. 35), have proved to remain stable during the migration, Brexit, COVID-19, Ukraine, and inflation polycrisis (Van Hecke et al. 2023).

While the communicative settings and mechanisms of political events such as EU summits have been studied rather extensively (Georgakakis and Rowell 2013; Beauvallet and Michon 2013; Huber 2012; Surubaru 2010; Cornia et al. 2008; Peter and de Vreese 2004), the regular interactions of Brussels-based EU politicians and correspondents and their communicative output are hardly the object of particular interest (Brüggemann et al. 2009, p. 397–98). Dedicated projects such as Europub and AIM have dealt with the role and attitudes of EU correspondents (Firmstone 2004; Statham 2004; Koopmans 2003; AIM Research Consortium 2007a, 2007b; Cornia et al. 2007; Hahn et al. 2006; De Bens et al. 2006) and there have not been any comparable projects in recent years. Some studies however attempted to dive into the communicative attitudes of MEPs (Oispuu 2011; Offerhaus 2010; Meyer 2009; Stamm 2006), Commission (Melchior 2017; Bauer and Ege 2012; Hooghe 2010), and Council officials (O'Reilly 2018; Savin 2011; Huster 2008; Schmidt 2008; Lewis 2007; Risse 2004). However, in recent years only a few scattered studies have added to this older body of literature (for an overview, see Michailidou and Trenz 2021).

All these studies fall short of systematically analysing the basic constellations and mechanisms behind the production of political communication output in the media. Furthermore, they hardly integrate the macro-level influences on the micro-level attitudes and behaviour. It has become quite clear that the EU's communication deficit is not merely the result of individual actors' personal intentions but has its origins in what Seoane Pérez (Seoane Pérez 2013, p. 102) calls a "twin deficit of domesticisation (lack of identity) and politicisation (lack of agonistic conflict)". In his interpretation, "the absence of a European demos and the curious mixture of neocorporatism, functionalism and diplomatic rule" (Seoane Pérez 2013, p. 102) are the true and systemic reasons for the communicative and democratic problems of the EU. Studies applying the principal-agent approach have demonstrated how these constellations guide the interests of political constituencies and media outlets as well as the individuals serving their respective electorates and audiences with their concrete communicative behaviour (Napoli 1997, p. 209).

Communication culture studies integrate these approaches and influences. Instead of observing communicative routines, they focus on actors' attitudes towards the production of political messages, the strategies behind it, and the underlying communicative rationality (Esmark and Mayerhöffer 2014). Political communication is understood as a process of exchange and cooperation between politicians and journalists within the specific configurations of the political communication system (Pfetsch 2014b, p. 18; Balcytiene and Vinciuniene 2009, p. 147; Pfetsch 2003, p. 203). What really shapes political EU communication output are "the empirically observable orientations of actors in the system of production of political messages toward specific objects of political communication, which determine the manner in which political actors and media actors communicate in relation to their common political public" (Pfetsch 2004, p. 348). Extensive comparative research within Europe has shown that these orientations differ between specific national or regional communication systems and explains these differences with exactly those system-level configurations that have been internalised by individuals through their organisational affiliation and socialisation (Pfetsch et al. 2014).

## 3. Research Framework

The political communication culture approach assumes that actors' attitudes are key in describing and explaining the communicative output of a political communication system. Through socialisation processes, these individual attitudes are shaped by systemic configurations within this communication system (Pfetsch 2014b). Linking macro and micro levels of political communication, individuals' or groups' attitudes can be evaluated against the systemic configurations that shape them. Accordingly, we want to find out what common or diverging attitudes guide EU politicians and journalists in Brussels during the process of political communication and what systemic configurations shape these attitudinal foundations.

The political communication culture concept has been used to describe different communication cultures on a national (Pfetsch 2014b; Pfetsch and Voltmer 2012; Pfetsch and Mayerhöffer 2011; Pfetsch 2003) and local level (Baugut et al. 2017). It gains relevance from its comparative approach and its ability to integrate systemic (macro level) influences and individual (micro level) attitudes. It has so far not been applied to transnational communication environments such as the politicians-journalists elite milieu in Brussels. This "methodological nationalism" (Wimmer and Glick Schiller 2002) is partly for conceptual reasons: It is easier to compare politicians' and journalists' attitudes in different countries and use these countries as separate and distinct units to explain structural differences than to acknowledge and integrate the multitude of overlapping micro- and macro-level influences that shape the communication culture in an elitist and international milieu as it can be found in the European quarter in Brussels.

We understand "Brussels" as a singular internationalised communication environment that has the potential to shape a distinctly "European" communication culture while acknowledging that "national political systems continue to provide the political and legal framework within which professional news journalists operate" (Michailidou and Trenz 2021, p. 4) and where politicians gain their legitimation. The Brussels communication environment is shaped by a duality of the national and the transnational: EU politicians and journalists have been socialised within specifically national communication cultures and distinct levels of professional distance and they orientate their communication efforts towards their mostly national electorates and audiences. However, EU politics has, by definition, a transnational dimension that EU politicians must consider when making decisions; there are highly relevant pan-European news outlets such as Politico, and the common communication context in Brussels must be expected to integrate national and professional differences among politicians and journalists at least to a certain extent (Mancini et al. 2007, p. 127). We, therefore, expect the political communication culture in Brussels to be an expression of the partly national, partly transnational character of the political and communication system and its inherent principal-agent relationships.

However, it remains unclear just how this communication culture can exactly be characterised. To answer this question, we use a heuristic that structures our analysis by defining one horizontal and one vertical dimension, each of them guiding our empirical examination. For each of the two dimensions, we compare EU politicians' and journalists' attitudes, searching for commonalities and differences between the two groups as well as between sub-groups defined by political institution (politicians) and media type (journalists). This way, we capture the potential constellations of communication culture in Brussels influenced by systemic as well as principal-agent-constellations and make them comparable with other communication cultures (Table 1). We integrate the actor group's professional attitudes (horizontal dimension) and national differences resulting from macro-level configurations (vertical dimension).

**Table 1.** Possible configurations of political communication culture in Brussels.

|  | **Strong Differences Politicians-Journalists (Distance)** | **Weak Differences Politicians-Journalists (Proximity)** |
|---|---|---|
| **Strong national influences (Nationalisation)** | segmented (1) | parochial (2) |
| **Weak national influences (Europeanisation)** | professional (3) | elitist (4) |

The horizontal dimension is what Baugut (2017, p. 52) has conceptualised as a continuum that places politicians' and journalists' attitudes between professional proximity or distance. Proximity means that politicians and journalists share the same attitudes while distance points towards a highly conflictual configuration, i.e., a deep divide between politicians' and journalists' attitudes. Due to the multi-segmentation of European public spheres, we add a vertical dimension that places the communicative output on the national vs. European continuum (Georgakakis 2013, p. 230; Preston 2009, p. 126). While attitudinal differences may appear due to different national communication systems within Europe that influence the actors in Brussels (Hallin and Mancini 2004) and diverging role images of politicians and journalists within those systems (Schwab Cammarano and Díez Medrano 2014), it would also be plausible that certain common attitudes arise because of the highly Europeanised communication context in which the actors are being socialised when working in Brussels.

If we integrate the two dimensions, four idealised types of EU political communication culture appear. In the case of a segmented political communication culture (1), strong national influences would complement a conflictual relationship between politicians and journalists. Brussels would merely be another battlefield for politicians and journalists as they find it in their respective home countries and the distinct national particularities appear in Brussels as well. In the second scenario (2), parochial communication culture, we would find strong national influences but hardly any remarkable conflicts between the professional groups. Politicians and journalists would more or less have the same attitudes towards political communication output but with strong differences shaped by national particularities. This scenario is rather hypothetical because it would involve approximation on the professional level while national differences would not be "overwritten" in the common communication environment in Brussels. In a professional communication culture (3), politicians and journalists maintain their distinct attitudes deriving from different roles in the political communication system, yet in the light of a Europeanisation of attitudes which erases national differences. If, in contrast, politicians and journalists understand themselves as part of one single elite group and tend to "overwrite" their initial socialisation in national communication systems with common European traits, then one could speak of an elitist political communication culture (4).

We understand this heuristic not only as a description of possible configurations of our dependent variable, i.e., the attitudes of Brussels-based EU politicians and journalists

towards the political communication output and therefore the communication culture in Brussels. It also guides our empirical analysis which looks for differences and communalities of politicians' and journalists' attitudes. The result of this analysis is not only hard to predict but also highly relevant because it helps to understand the consequences of the existing macro- and meso-level configurations in EU politics and journalism and its interplay. If, say, an elitist mode of cooperation between politicians and journalists was to be observed, this might point towards the potential for further Europeanisation of political EU communication, but also a rather uncritical way of reporting that might raise scepticism and reduce the trust people place in European democracy.

We structure our analysis around three research questions. We assess

(1) What strategies are perceived to be effective,
(2) How EU politicians and journalists evaluate these strategies with regard to their day-to-day experiences and their audience, and
(3) How frequently actors with different audiences and constituencies interact in Brussels.

Analysing these three aspects enables us to understand whether and how strongly the Brussels communication environment integrates the different communication cultures and principal-agent relationships into the basic attitudes and mechanisms of political EU communication. Showing commonalities or differences among EU politicians and journalists will show whether the transnational and unifying influences of the communication context are stronger or the differences between the professional and national groups.

The first research question refers to the strategies that politicians and journalists apply to raise attention to their topics and views.

*RQ1: What output strategies are seen as effective?*

It is particularly important to understand what strategies determine the way that political communication output is mutually created and, in a second step, mediated to European audiences—and whether there is consensus among politicians and journalists about their efficiency or not. The question in the questionnaire was "Politicians may use various ways to get public attention. In your opinion, how effective is . . .", followed by a list of output strategies. These strategies can then be clustered along established groups such as Frontstage and Backstage (Pfetsch et al. 2014).

Because of the strong tendency to report EU politics from a national angle, we analyse this national framing separately, asking correspondents to rate their approval of the statement "If you want people to comprehend EU politics, you have to put a national spin on communication about it". Older studies have shown that communicating EU politics with a national twist is understood as instrumental in order to fulfil the information needs of the audience as well as the expectations of media outlets (Mancini et al. 2007, p. 135; Schudson 2003, p. 47) and connects to the idea of "translating" complex political procedures and decisions (Schmidt 2008, p. 118; Heikkilä and Kunelius 2007, p. 29; Gleissner and de Vreese 2005, p. 238).

*RQ2a: How do politicians and journalists assess the adequacy of the output strategies?*
*RQ2b: How do politicians and journalists assess the knowledge of their common audience?*

Our analysis also takes the motivations behind actors' attitudes towards these output strategies into account: Do they use them because they think it is the right thing to do or rather for opportunistic reasons? This aspect is important if we want to understand the interplay of national communication cultures and audience/constituency orientations on the one hand and the perspectives that EU politicians and journalists possibly gain when working in Brussels and communicating with colleagues from other countries. While these "European" perspectives may not directly change the way EU politics is being communicated because the actors need to predominantly fulfil their agent role, it may point out that dominance of national frames and perspectives is not a given thing within Europe.

We complement this aspect with actors' views towards their common audience. It is no secret that European citizens have low levels of interest in and knowledge about

EU politics (Gattermann and Vasilopoulou 2015). Furthermore, earlier studies of political communication cultures indicate that "elite politicians and elite journalists in their complex and sometimes traumatic internal relations distance themselves from the people whom the one group is supposed to represent and the other group is opposed to serve" (Moring and Pfetsch 2014, p. 300). Even more so than communication elites in national systems, actors in Brussels appear to be prone to understand themselves as a distanced elite with little obligation to serve their distant and uninterested home audiences (Mancini et al. 2007, p. 126).

*RQ3: How are contact networks in Brussels configured, i.e., which actor (sub)groups work particularly closely together in order to create political communication output?*

The third research question aims at explaining the political communication output by analysing the actors' contact networks. Political communication culture conceptualises communication output as a direct consequence of such politicians-journalists interactions oriented towards a common audience. We therefore understand these networks as proxies of the macro-level features of the political communication environment in Brussels, namely the intensity of international cooperation among politicians and journalists in the light of a segmentation of European publics and audiences with their specific demands on reporting EU politics.

From an audience perspective as well as in a principal-agent model, these needs are catered for by different media types with different target groups. From a voter's perspective, some EU institutions and their actors cater more directly to their national electorates (e.g., national governments in the Council) and some are more focused on common European interests (e.g., the Commission). Working for one distinct media outlet or EU institution inside the Brussels communication context will not only shape the attitudes and actions of politicians and journalists but it is also highly plausible that politicians and journalists with overlapping audiences have more contact to work on political communication than those with completely separated audiences.

This is relevant to our analysis. We expect intra-group differences, particularly among politicians and journalists due to their distinct audiences or electorates. Most EU politicians and journalists either have a more regional or national audience or constituency (e.g., regional public service broadcasters or nationwide newspapers, Members of the European Parliament or Permanent Representatives) or a rather international orientation (e.g., elite media such as the Financial Times and Politico or outlets specialising in EU politics).

In the next two chapters, we describe and compare the attitudes expressed by 309 politicians and journalists in Brussels for each of the three research questions.

## 4. Data, Measures and Analysis

The empirical study draws on a survey conducted from May to September 2016 among 309 EU politicians and correspondents based in Brussels. We produced a dataset that is unique in breadth and depth. While we acknowledge possible criticism that the data appears to be outdated, we believe that it does not really stick. The basic configurations of the Brussels communication environment have proven to be very stable over time, including, among others, the institutional political setup (Hall and Mérand 2019), the structure of the media and their correspondents (Harding 2016), and communication routines (Melchior 2017), even during times of crisis. We therefore expect the communication culture in Brussels to be very stable and believe that the Brussels machinery kept producing political communication output in the same way as in 2016 since the polycrisis of migration, COVID-19, the Ukraine war and inflation arose. Even Eurosceptic populists appear to adapt to the traditional communication routines (Plavec 2020, p. 153ff). Our data manifest the systemic structure of the elite milieu in Brussels, which has become more important and proved more stable during the polycrisis. We therefore believe that our data, even though being collected during a more "settled" period, are still relevant.

A few restrictions on the representativeness were inevitable due to the limited availability of data. This is particularly true for the random selection of the participants. Instead, a positional approach has been used for the selection of politicians and top-level civil servants in the European Commission (Commissioners and deputy Commissioners, Directors-General), high-level diplomats in the European Council (three national representatives for each member state), and the European Parliament (heads of political groups, committees, and national delegations). This approach expects that actors in higher ranks are more influential than those in lower ranks (Maurer and Vähämaa 2014). Due to a lack of a comprehensive and up-to-date official list of all accredited EU correspondents, journalists were selected using (partly) older lists provided by the European Commission and the permanent representations of the member states with the precondition that all of them work permanently in Brussels. All journalists on these lists fulfilling the requirement were contacted and invited to participate in the survey.

Politicians and journalists received paper questionnaires as well as follow-up emails with links to an online version of the questionnaire. In total, 140 politicians and 169 journalists responded (utilisation rate: 33.3% for politicians and 34.0% for journalists). Following the classification of Hallin and Mancini (Hallin and Mancini 2004, 2012) and further, more detailed work of Castles and Obinger (2008, pp. 336–38), Jakubowicz (2012) and Moring and Pfetsch (2014), politicians and journalists were coded as part as one of five country groups (Appendix A). There were 21 participants from the Anglophonic EU member states, 65 from Eastern Europe, 88 from the German-speaking countries (including, for reasons of similar communication systems, the Netherlands and Belgium), 28 from Northern Europe/Scandinavia, and 79 from Southern Europe. 28 participants gave no information on their nationality. The composition of the country groups is documented in the annex.

Because of the lack of up-to-date information on the exact number of journalists from the respective member states, the representability of the data can only be estimated. Actors from the Southern European member states seem slightly overrepresented, accounting for 35.9% of the participants while making up only 25.9% on the lists. For all other actor groups, the difference is lower than 5 percentage points. We also checked the representativeness according to media type, i.e., journalists working for newspapers, public service or private broadcasters, and online media. No such media type is over- or underrepresented by more than 5 percentage points. Among politicians, response rates in the permanent representations were particularly higher (52% compared to 30% and 28% in the Commission and Parliament). While this limits the potential for fined-grained analyses on a subgroup level, the sample does enable us to reasonably compare the attitudes of actors from professional and country groups.

## 5. Findings

*RQ1: What output strategies are seen as effective?*

Our first step to understanding the output dimension of political communication culture in Brussels is to find out which output strategies appear most effective to politicians and journalists to make their political stance visible to their common audience and to influence the political agenda. As our analysis focuses on actors' attitudes towards political communication output, we operationalise the dependent variable of RQ1 as the perceived effectiveness of eight output strategies to get the attention of their common audience for certain topics or messages. We used items developed by Donges et al. (2014) and Esmark and Mayerhöffer (2014) but adjusted them to the Brussels communication context.

Politicians and journalists had to evaluate the effectiveness of eight output strategies: a speech in the European Parliament, a press release, a Facebook post, leaking stories to selected journalists, gearing stories towards conflict or drama, appearing on a TV talk show, an interview in the Financial Times and an appearance in a national newspaper. Replicating a factor analysis by Pfetsch et al. (2014), we identify three different strategies (Table 2): Frontstage strategies where a politician mostly speaks for him- or herself,

Backstage strategies focusing on journalistic news values and strategies that use media platforms as multipliers for messages.

**Table 2.** Brussels communication elites identify three main ways to influence the political agenda (Factor analysis for six output strategies, N = 309).

| Output Strategy (Load) | Factor |
|---|---|
| Speech in the European Parliament (0.847)<br>Press Release (0.831) | Frontstage |
| Leaking information to selected journalists (0.807)<br>Gearing stories towards conflict and drama (0.778) | Backstage |
| Appearance on a TV talk show (0.836)<br>Appearance in a national newspaper (0.821) | Multiplier |

Extraction method: Main component analysis, rotation method: Varimax with Kaiser normalisation, Eigenwert > 1, all items loading with at least 0.7 on the respective factor.

Generally, EU politicians and journalists perceive frontstage strategies to be far less effective than backstage or multiplier strategies (Figure 1). Only 27% of the actors participating in the study ranked a speech in the European Parliament or a press release as effective or very effective (backstage strategies: 58%, multiplier strategies: 59%).

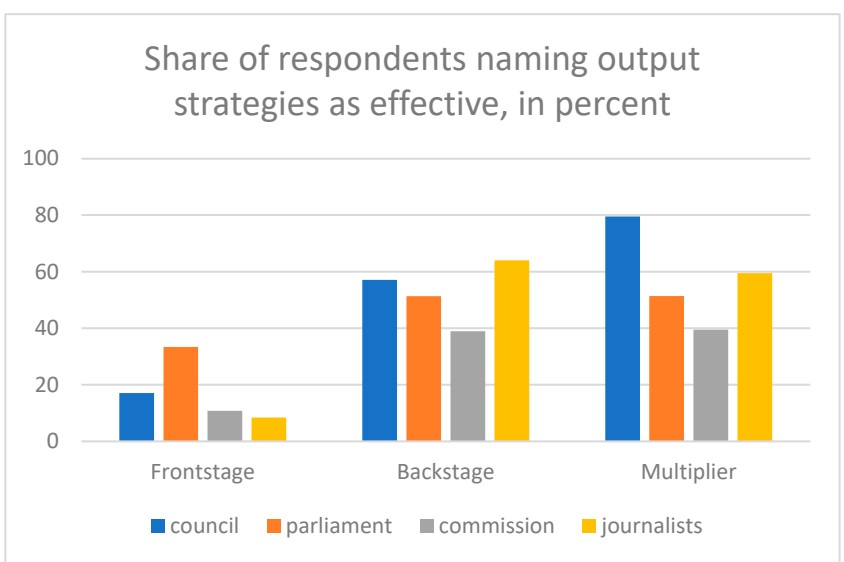

**Figure 1.** Effectiveness of output strategies.

However, there are differences between the actor groups and sub-groups: Members of the European Parliament (33%) rate frontstage strategies more effective than journalists (8%). Commission officials and MEPs rate the effectiveness of backstage and multiplier strategies lower (51%/39%) than journalists (64%/60%). The level of perceived effectiveness for multiplier strategies is highest among politicians in the European Council (80%) and lowest among Commission officials (40%). Through a principal-agent lens, this makes sense since the Council group consists mainly of member states' permanent representatives with a strong orientation towards their home country and hints at different communication strategies among the more supranational-oriented Commission officials.

Differences along national lines are only significant for frontstage and backstage strategies. While frontstage strategies seem to be relatively popular among actors from Eastern European states (38% vs. 27% overall), backstage strategies are most popular among Northern European actors (76% vs. 58% overall). We interpret this as an expression of different national media cultures and traditions that remain influential in the internationalised Brussels communication environment.

We conducted a separate analysis for the output strategy of national framing, i.e., a "translation" of EU politics for national audiences by applying a national spin to political messages. Firmstone (2004) describes this strategy as very popular among Brussels actors, but it has been criticised as perpetuating national perspectives on European politics and therefore hindering a Europeanisation of political discourses (Statham 2008). Rather unsurprisingly, national framing of EU messages is almost universally accepted as an effective strategy in communicating EU politics, with 75% of all actors naming it as effective or very effective with hardly any professional or national differences.

Summarising the results for RQ1 for our heuristic, the basic attitudes of the politicians and journalists towards effective output strategies are largely Europeanised—there are no strong national or regional differences. While some professional distance between politicians and journalists is observed for the three clusters of output strategies, there is almost complete unanimity towards the strategy of national framing. Pointing out the relevance for national audiences and reporting through a national is consensually perceived as an effective output strategy.

*RQ2a and RQ2b: Adequacy of output strategies and image of the audience.*

How do political communication elites assess output strategies besides their effectiveness? For example, do they use a national frame merely because it is most efficient or because they think it is the right thing to do? This question relates to the original idea of understanding communication culture as a result of socialisation within a communicative context. Socialisation can be "weak" or "strong" (Checkel 2007, p. 6). In a "strong" socialisation scenario, actors have internalised the norms that govern their actions, "weak" socialisation leads to a merely strategic adoption of norms. This may particularly be the case for the EU politicians and journalists in Brussels who work in a highly elite and cosmopolitan environment and communicate supranational politics—but communicate about it for a seemingly remote audience in their home country.

EU politicians and journalists were asked whether it is in their view essential to understand and reflect on political positions in other countries if one wants to communicate EU politics appropriately (RQ2a) and whether ordinary citizens do or do not really understand EU politics (RQ2b). The results are almost unanimous among actor and country groups: reflecting other national positions on European politics is thought to be essential (81% acceptance), while 71% of the politicians and journalists participating in the study have a negative conception of citizens' ability to understand EU politics.

The consensus that any communication output that focuses on merely national perspectives is neither adequate nor sufficient is complemented by the almost unanimous assessment that there is very little understanding of EU politics among the common audience(s). It appears that politicians and journalists intentionally and against their true convictions choose to push political messages towards a national angle—because they assume citizens otherwise would not get the message.

With our heuristic in mind, the answers to RQ2.1 and RQ2.2 clearly point toward the elitist model of political communication culture: Professional or national differences among politicians and journalists are almost completely absent.

*RQ3: Contact networks as defining structures of political communication output.*

Contact networks are vitally shaping the political communication output—particularly in a place like Brussels that is flooded with official information but where relevant information is often shared with selected actors (Martins et al. 2012). Contact networks are generally seen as "an integral factor for the sourcing of information for Europeanised EU-journalists" (Kümpers 2022, p. 13), particularly in addition to official sources that may be shaped by national networks in Brussels or a correspondent's audience. As a consequence of the distinct communication context in Brussels, EU institutions traditionally favour correspondents with a transnational audience (Statham 2008, p. 406). Are contact networks in Brussels generally rather national or supranational, or does it all depend on the audience and electorate?

When asking journalists, audience is key. Journalists whose media outlets address an international audience said that they meet people from countries other than their own home country much more often. Asked about the frequency of contact with politicians from their home country vs. other countries on a 1–5 scale (1 = never, 5 = daily), journalists working for media with an international audience report up to 0.9 points more contacts with politicians from other than their home country—with the same difference for journalists with a national audience for politicians from their home country.

For EU politicians, it is more difficult to define the national or international character of their audience or electorate (Table 3). Council officials and Members of Parliament have the strongest contact intensity for newspaper and broadcast journalists from their home country. Commission officials appear to have the lowest contact intensity with journalists, no matter what type of medium they report for. This does not have to mean that the Commission does not make its point in public—it does, in fact, mostly communicate via its spokespersons who were not part of this study. Nevertheless, comparing contact intensity with different groups of journalists and, indirectly, different national and international audiences shows that contact networks in Brussels appear to be particularly strong among fellow countrymen.

**Table 3.** Differences in contact intensity with politicians from journalists' home country and other countries, by audience orientation of journalists' media outlet (average value, 1 to 5).

| Politicians Subgroup | International | European | National | Regional |
|---|---|---|---|---|
| Council officials | −0.2 | −0.1 | +0.8 | +0.6 |
| Commissioners | −0.9 | −0.5 | +0.3 | +0.3 |
| Members of European Parliament | −0.3 | −0.1 | −0.2 | −0.4 |
| N | 24 | 37 | 119 | 20 |

Our results indicate that despite many opportunities to meet politicians and journalists from other countries, contact patterns within the Brussels milieu of communication elites are strongly defined by national networks, at least for a majority of the actors whose attitudes we analysed. For politicians, there is a diffuse proximity to journalists from their home country. Within the journalists, we see a strong orientation of contact networks along with the audience orientation of the different media.

We have conceptualised contact networks as proxies of the EU communication system and its derived principal-agent relationships. Our results confirm this understanding. Journalists working within national media systems, which are still more or less confined to nationally defined audiences, seem to have little interest in meeting politicians from outside their home country. This is, however, not the case for journalists who address international, mostly elite audiences. For politicians' contacts with journalists, we interpret our results as a consequence of partly overlapping, partly conflicting orientations towards national electorates on the one hand (e.g., MEPs with their constituency) and supranational policy orientations or, more generally, a responsibility towards all European citizens—an ambiguous role that is a consequence of the political system of the EU.

The results for RQ3 show that there is a differentiation along national or regional boundaries but that contact behaviour somehow "matches" between professional groups according to their national vs. transnational orientation. This points towards a rather "parochial" way of making contacts in Brussels.

## 6. Discussion and Conclusions

Our study provides evidence that political communication output in Brussels is predominantly driven by the cooperation of politicians and journalists following a "media logic" (Altheide and Snow 1979), adhering to journalistic assessments of newsworthiness and production routines. Political communication output from Brussels is shaped by

a seemingly cynical conception of the European audience(s) and cooperation between politicians and journalists mainly takes part within national contact networks.

These findings clearly confirm a strong orientation towards a national spin of political messages, "translating" EU politics into messages and frames that fit into national communication patterns and traditions (Michailidou and Trenz 2021, p. 9). Our study indicates that this tradition, while entirely explainable from an actor's perspective, is a major driver of the EU's communication deficit. That is ironic because EU politicians and journalists themselves do not even believe that this communication strategy is adequate for a complex and in many aspects truly transnational policy field such as EU politics—an aspect that our study adds to the picture for the first time. On top comes a sceptical view of their audiences that the elites share regarding any change in political communication, particularly one into a more Europeanised way of formulating messages.

How can the production of political communication output in Brussels be located within the two-dimensional heuristic outlined above? Concerning the effectiveness of certain output strategies, we observe a moderate professional distance between politicians and journalists, particularly relating to frontstage output strategies that are only popular among a minority of politicians. Besides that, hardly any peculiarities appear in our results that could be explained by differences between national media systems or communication cultures. Beliefs in the effectiveness (and deficits) of the popular "national framing" strategy are almost unanimous among the Brussels communication elites, as well as deeply rooted mistrust in the audience's capacity to really understand EU politics. Contact networks, though, seem to be partly nationalised, reflecting a segmentation of European audiences and electorates (Table 4).

**Table 4.** Political communication culture in Brussels (output dimension).

|  | **Distance** |  | **Proximity** |
|---|---|---|---|
| **Nationalisation** | segmented |  | parochial |
|  |  |  | contact networks |
| **Europeanisation** | professional | output strategies | national framing |
|  |  |  | reflect other positions |
|  |  |  | negative image of audience |
|  |  |  | elitist |

Our analysis finds hints for parochial, professional, and elitist aspects of political communication culture concerning the communication output coming from Brussels. While no clear pattern appears in our analysis, political communication output in Brussels seems to be leaning towards the elitist model. Professional differences, if any, only tend to become relevant when assessing the effectiveness of certain output strategies while there is common support for other strategies from both politicians and journalists. Rather strong national default lines appear when analysing contact networks and the intensity of the communication elites. It must however be noted that hardly any journalist or politician in Brussels can work entirely without contact with international colleagues or sources.

Our study shows that the macro- and meso-level constellations of EU politics and political communication express themselves in the attitudes of EU politicians and journalists, particularly in their contact networks and output strategies with their strong focus on national framing. While this is not very surprising, we also read the results as an indicator that the Europeanised communication environment in Brussels leads to a diminution of professional and national differences among politicians and journalists. That hints towards some Europeanisation of the politicians' and journalists' attitudes towards political communication output and is a strong indicator of the unifying influences of the internationalised Brussels communication environment. From a principal-agent perspective, this points towards some divergence of the principal's and agent's interests and loyalties and represents a control problem—at least if one does not acknowledge at least some interest of the principals in a federalisation of EU politics as it may be the case during

times of crises. From a more macro-oriented political communication culture standpoint, a harmonisation of EU politicians' and journalists' attitudes is easily explicable as a result of the need to find a common basis for (political and communicative) business ("gemeinsame Geschäftsgrundlage", Pfetsch 2003, p. 131).

While the proponents of a further Europeanisation of political EU communication may welcome this finding, another result of our study is rather alerting. The cynic elitism that is shared by a large majority of our respondents will certainly not help if the goal is to bring the EU closer to the citizens and can be understood as a consequence of Seoane Pérez (2013, p. 160) assessment that "the EU is the most democratic of all international organisations, but the popular control of the representatives is weak and indirect".

If we understand contact networks as a consequence of the systemic configurations of European public spheres, media, and political systems that become mediated through organisations and principal-agent relationships, there appears little room for change. This would require the actors to be convinced that a different way of communicating EU politics is a better way to reach their goals. It seems obvious that politicians with a national or regional electorate as well as journalists reporting for media with a national or regional outreach are incentivised to focus their communication efforts on aspects of EU politics that have a direct impact on their voters or readers.

Our study sheds new light on the mechanisms behind the EU's communication deficit. It confirms that this deficit is in part a consequence of the fragmentation of European publics and constituencies. However, it also makes clear that besides these systemic configurations, the communication environment in Brussels shapes the attitudes in the elite milieu of EU politicians and journalists. The way that these actors cooperate and communicate towards their common public(s) is not merely the result of what their voters or media outlets want them to do but rather shaped by the specific character of this microcosm itself—with consequences for political communication inside the whole EU. The EU's political and communication system may remain fragmented along national boundaries, but that does not stop the politicians and journalists representing these countries from mingling in Brussels on a regular basis and developing shared views and attitudes towards political communication that transcend these very boundaries.

While it becomes clear that the result is a tendency towards an elitist and cynical communication culture among the actors in Brussels that most probably will not be able to overcome the macro-level shortcomings leading to a communication deficit of the EU, no clear way out can be drawn based on the results. This paper needs to be understood as rather exploratory within an under-researched section of EU political communication studies. Another caveat is that our results may be hard to generalise in other transnational communication contexts because of the particular systemic structures, incentives, and constraints of the Brussels communication environment.

Future studies should analyse more in detail how strongly national routines and socialisation influence communicative behaviour in Brussels. Other results of the survey presented here confirm that there is a tendency among actors in Brussels to develop common attitudes and thereby forming a distinct elite milieu (Plavec 2020). Apart from the socialisation mechanisms behind this harmonisation, it would be worth the effort to take a closer look at how this socialisation translates into communicative attitudes and behaviour. On a general level, the results we presented here are a clear lead that an actor- and attitude-centred approach promises more insights concerning the processes and configurations behind the shortcomings of political EU communication.

**Author Contributions:** Conceptualization, J.G.P. and B.P.; methodology, J.G.P. and B.P.; software, J.G.P.; validation, B.P.; formal analysis, J.G.P.; investigation, J.G.P.; resources, J.G.P.; data curation, J.G.P.; writing—original draft preparation, J.G.P.; writing—review and editing, J.G.P. and B.P.; visualization, J.G.P.; supervision, B.P. All authors have read and agreed to the published version of the manuscript.

**Funding:** The publication of this article was funded by Freie Universität Berlin.

**Informed Consent Statement:** Informed consent was obtained from all subjects involved in the study.

**Data Availability Statement:** Data has to remain private due to confidentiality of the participating politicians and journalists.

**Conflicts of Interest:** The authors declare no conflict of interest.

## Appendix A  Country Groups

| |
| --- |
| Anglophonic |
| United Kingdom |
| Ireland |
| Eastern Europe |
| Bulgaria |
| Croatia |
| Czech Republic |
| Estonia |
| Hungary |
| Latvia |
| Lithuania |
| Poland |
| Romania |
| Slovakia |
| Slovenia |
| German-speaking |
| Austria |
| Belgium |
| Germany |
| Luxemburg |
| Netherlands |
| Northern Europe |
| Denmark |
| Finland |
| Sweden |
| Southern Europe |
| Cyprus |
| France |
| Greece |
| Italy |
| Malta |
| Portugal |
| Spain |

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
