# Peer review of "Europeanised Attitudes, Nationalised Communication? Evidence on the Patterns behind Political Communication Output in Brussels"

_journalmedia, doi:10.3390/journalmedia4030062_

Round 1
Reviewer 1 Report
This submission touches upon an issue that, from communications and political meaning-making point of view, calls for being defined as an issue of heightened scholarly relevance but also complexity.
This is due not only to the complex political-thematic issues addressed at the level of European institutions, which require strategic policymaking on macro level but also to various characteristics such as trends and traditions in national reporting and political communication. These characteristics are further defined by the specificalities of the “Brussels culture”, i.e. the Brussels microcosm, and they act as critical factors determining how the communications output is produced to reach national publics.
Generally, the scholarly attention has been highly focused on the Brussels context some 15-20 years ago, particularly at the time when the problems of communications deficit and the evolution of the European public sphere through institutional and cultural integration were extensively debated and explored by various analysts. Related issues have also been attracting heightened scholarly and political attention. The time was ripe then to discuss communication deficit, dissociations between journalistic traditions and national political parallelisms, hence the concept of communications deficit seemed valid at that time. I attentively follow the authors’ proposal to apply the same concept as a term that refers to European communicative matters of enduring character. Likewise, I would like to congratulate the decision of the authors to revisit this thematic field by delving deeper in the analysis of how national communicative outputs are being defined by the specificities of routines and communications culture in Brussels. However, this ambition has been only approximated and fully realized.
To minimize this feeling of incompleteness, I suggest radical revision within the Introductory section of this article to highlight the specificities of the research focus to disclose Europeanization of attitudes among politicians and journalists, which is to be tackled here. Moreover, the contextualization of the chosen “research problem” – whether defined as (a) the specificities of communicative deficit on multiple levels and factors (internal information management and cultural needs, or “populist shift” in politics) determining such an outcome, or (b) the specificities of European communication in times of “epistemic crises”, shaped by changed demands resulting from the socio-political transformations and uncertainty, which was brought in relation to the Covid pandemic, and also by the changed geo-political landscape and socio-economic insecurity in Europe as well as globally due to the Russian aggression and war in Ukraine, as well as by socio-technological specificities defining structures and professional systems of content production and communication – remains of utmost significance in this article.
Furthermore, the specificities of the research problem and its context must be revisited and highlighted once more in the Discussion section, and all the results from the empirical analysis should be interpreted in reference to these contextual conditions.
In addition, the research focus on the roles of actors (journalists and politicians) as well as the formation of their attitudes requires taking into the account another theoretical concept, namely that of “agency”. The specific characteristics of individual agency are shaped by professional norms and routines. Hence, it is at this stage where theoretical (conceptual) and practical (empirical) aspects of the analysis in this article can come into a coordinated manner. Thus, I suggest adding this aspect of agency into the Literature Review section. Without such a connecting line, no adequate linkages could be seen between the conceptual model and data collection and analysis.
To conclude, the major drawback I observe in this study is not directly the choice of the data to be analyzed. Although the authors have made attempts to include some newer references and analyses, this does not compensate the conceptual weakness in this version of the paper. The data, collected some 9 years ago, would have produced valuable results in the “settled period” of communication. However, the past few years have brought an “unsettled period” with a number of crises, each of which has serious implications on the ways information and news is selected, produced, and shared.
At this moment, such relevance is missing in this article.
Author Response
We appreciate the reviewers' and editors' work and want to thank them for thoroughly discussing our article. The reviewers' comments were constructive for improving our work. We considered each comment carefully and changed the manuscript accordingly. We carefully revised the entire manuscript and highlighted the parts with the most profound changes in yellow. In the following, we respond in detail to the reviewers' comments.
Comment 1): This submission touches upon an issue that, from communications and political meaning-making point of view, calls for being defined as an issue of heightened scholarly relevance but also complexity.
This is due not only to the complex political-thematic issues addressed at the level of European institutions, which require strategic policymaking on macro level but also to various characteristics such as trends and traditions in national reporting and political communication. These characteristics are further defined by the specificalities of the “Brussels culture”, i.e. the Brussels microcosm, and they act as critical factors determining how the communications output is produced to reach national publics.
Response: Thank you for the overall positive evaluation of the research gap and contribution the paper tries to make. Given the complexity of the issue and our research setup, this feedback is very helpful to make our work as useful as possible.
We report on the changes made to the manuscript in detail below.
Comment 2): Generally, the scholarly attention has been highly focused on the Brussels context some 15-20 years ago, particularly at the time when the problems of communications deficit and the evolution of the European public sphere through institutional and cultural integration were extensively debated and explored by various analysts. Related issues have also been attracting heightened scholarly and political attention. The time was ripe then to discuss communication deficit, dissociations between journalistic traditions and national political parallelisms, hence the concept of communications deficit seemed valid at that time.
I attentively follow the authors’ proposal to apply the same concept as a term that refers to European communicative matters of enduring character. Likewise, I would like to congratulate the decision of the authors to revisit this thematic field by delving deeper in the analysis of how national communicative outputs are being defined by the specificities of routines and communications culture in Brussels. However, this ambition has been only approximated and fully realized.
Response: Thank you very much for your positive comments. We have revised the introductory section in this light and have, among others, added Seoane Pérez’ (wider) perspective on the communication deficit. We therefore hope to make the enduring relevance of this issue clearer.
Comment 3): To minimize this feeling of incompleteness, I suggest radical revision within the Introductory section of this article to highlight the specificities of the research focus to disclose Europeanization of attitudes among politicians and journalists, which is to be tackled here.
Response: We undertook the “radical revision” of the Introductory section and clarified the scope of our research. While the Europeanization of attitudes (vs national “pillarization” as observed in the European communication system) remains the main focus, we eliminated most of the normative tonality.
Comment 4): Moreover, the contextualization of the chosen “research problem” – whether defined as (a) the specificities of communicative deficit on multiple levels and factors (internal information management and cultural needs, or “populist shift” in politics) determining such an outcome, or (b) the specificities of European communication in times of “epistemic crises”, shaped by changed demands resulting from the socio-political transformations and uncertainty, which was brought in relation to the Covid pandemic, and also by the changed geo-political landscape and socio-economic insecurity in Europe as well as globally due to the Russian aggression and war in Ukraine, as well as by socio-technological specificities defining structures and professional systems of content production and communication – remains of utmost significance in this article.
[…]
To conclude, the major drawback I observe in this study is not directly the choice of the data to be analyzed. Although the authors have made attempts to include some newer references and analyses, this does not compensate the conceptual weakness in this version of the paper. The data, collected some 9 years ago, would have produced valuable results in the “settled period” of communication. However, the past few years have brought an “unsettled period” with a number of crises, each of which has serious implications on the ways information and news is selected, produced, and shared.
At this moment, such relevance is missing in this article.
Response: We extended and revised the introductory section and the literature review to tackle this issue. There is now a broader understanding of the communicative deficit in the article as it was laid out by Seoane Pérez (2013).
While the polycrisis presents a challenge to EU politics, we also believe that it has – like, historically, all of the EU’s crises – strengthened the institutional setup that shapes, beneath many other things, the communication culture in Brussels. We have elaborated this argument in the data section as well. The data, while being collected seven years ago, remain a unique insight into the Brussels microcosm which we believe has remained stable throughout the years of crisis.
There can be no doubt about the huge impact of the numerous crises named in the comment. The media system is under enormous pressure and in the middle of a historical transformation. However, the specific constellations of EU politicians and journalists producing political messages for EU citizens are relatively unharmed by these developments; the cooperation between EU politicians and political correspondents is a rather “traditional” business with the news media still being the prime source for information and commentary about EU politics.
For many other research projects, data from 2016 collected under the circumstances of a “settled” period could hardly be seen as a valuable source of new insights. In our case, we believe that this is still the case. We have laid out these arguments in the revised version of the paper and hope that along the other major revisions and improvements, the relevance for today’s EU political communication (deficit) becomes much clearer than in the previous version of the paper.
Comment 5): Furthermore, the specificities of the research problem and its context must be revisited and highlighted once more in the Discussion section, and all the results from the empirical analysis should be interpreted in reference to these contextual conditions.
Response: We have revised the discussion section and believe that it now picks up the contextual conditions (which have been added to the introduction) much better than before.
Comment 6): In addition, the research focus on the roles of actors (journalists and politicians) as well as the formation of their attitudes requires taking into the account another theoretical concept, namely that of “agency”. The specific characteristics of individual agency are shaped by professional norms and routines. Hence, it is at this stage where theoretical (conceptual) and practical (empirical) aspects of the analysis in this article can come into a coordinated manner. Thus, I suggest adding this aspect of agency into the Literature Review section. Without such a connecting line, no adequate linkages could be seen between the conceptual model and data collection and analysis.
Response: Thank you for this very important aspect. We have added references to the principal-agent concept in numerous parts of the paper. Indeed, our results are in some ways maybe not counterintuitive, but some kind of addition to how one would expect the principal-agent relationships in Brussels to turn out regarding the attitudes towards communication output.
We believe that the value of our paper is in part to show that communication routines are one thing and the attitudes of the actors are another thing. By adding the principal-agent concept, we make this a lot clearer and can much better elaborate the additional insights into the research problem that the communication culture approach can deliver.
Reviewer 2 Report
This study of the Brussels bubble, on the interaction of politicians and journalists and their political communication strategies, is relevant because it offers an elegant matrix to understand such relationship, with two axis (professionalism and europeanization) that lead to four possible outcomes.
In all this discussion it is striking that the authors do not cite Seoane Pérez's 'Political communication in Europe' book (Palgrave, 2013), as it offers a challenging interpretation of the EU's communication gap by providing an structural and cultural explanation. That is, in order to understand why EU political communication is the way it is, you need to look at how the EU was integrated, how it is governed, and what sort of political community the EU is.
The Europeanized public sphere is only an elite one, and it is integrated by EU actors themselves and their publics (interest groups), not the European public, which is still a fiction.
That little bubble aside, all EU communication is nationally mediated, as national is the remit of most democratic politics in the EU (no child dreams of being an EU president, access to EU political positions is nationally mediated, there's not a European political career, only a national with a European outpost). This systemic conditions need to be acknowledged.
Also, as Seoane Pérez argues, EU politics are filled with interests towards non-transparency (many EU and national actors are interested in making the EU a 'black box' and a space beyond politics, despite their lip-service towards full transparency), which might explain the authors' finding of that preference for backstage dealings between journalists and their sources.
Author Response
We appreciate the reviewers' and editors' work and want to thank them for thoroughly discussing our article. The reviewers' comments were constructive for improving our work. We considered each comment carefully and changed the manuscript accordingly. We carefully revised the entire manuscript and highlighted the parts with the most profound changes in yellow. In the following, we respond in detail to the reviewers' comments.
Comment 1): This study of the Brussels bubble, on the interaction of politicians and journalists and their political communication strategies, is relevant because it offers an elegant matrix to understand such relationship, with two axis (professionalism and europeanization) that lead to four possible outcomes.
Response: Thank you for this overall positive evaluation of our work and for your suggestions on improving the manuscript. We report on the changes made to the manuscript in detail below.
Comment 2): In all this discussion it is striking that the authors do not cite Seoane Pérez's 'Political communication in Europe' book (Palgrave, 2013), as it offers a challenging interpretation of the EU's communication gap by providing an structural and cultural explanation. That is, in order to understand why EU political communication is the way it is, you need to look at how the EU was integrated, how it is governed, and what sort of political community the EU is.
Response: We are fully aware of Seoane Pérez enlightening comments to and clarifications of the EU’s communication deficit. They are valuable for the scope of our study and the interpretation of our results. We have therefore added references to his work in the introduction, literature review and discussion sections of the paper.
Comment 3): The Europeanized public sphere is only an elite one, and it is integrated by EU actors themselves and their publics (interest groups), not the European public, which is still a fiction.
That little bubble aside, all EU communication is nationally mediated, as national is the remit of most democratic politics in the EU (no child dreams of being an EU president, access to EU political positions is nationally mediated, there's not a European political career, only a national with a European outpost). This systemic conditions need to be acknowledged.
Response: Thank you very much for this comment. We see a duality of very strong national configurations within the European public sphere(s) but also the highly internationalized communication environment in Brussels. The relevance of analyzing the elite milieu results from their multiplicator role through access to the main media channels where EU news reach European citizens. However, we made the national influences very clear in our revision of chapter 3 (research framework).
Comment 4): Also, as Seoane Pérez argues, EU politics are filled with interests towards non-transparency (many EU and national actors are interested in making the EU a 'black box' and a space beyond politics, despite their lip-service towards full transparency), which might explain the authors' finding of that preference for backstage dealings between journalists and their sources.
Response: Thank you for this reference. We see some plausibility in Seoane Pérez’ argument, though we also see that there has been quite some politicization of EU politics in the recent years during the polycrisis with its intensification of political activity and intervention on EU level. We interpret the actors’ preference for backstage strategies rather as a hint towards a professionalization. EU politicians and journalists simply seem to acknowledge that “backstage” channels adapt better to the configurations of the audience’s information behavior.
Round 2
Reviewer 1 Report
Thank you for the time devoted to updates and explanations; the only remaining comment which I would add in the final assessment is the invitation to the authors to conceptually define the idea of a "mindset" as an attitudinal and human-centric aspect required in communication analysis.
Author Response
Thank you for this valuable comment! We have used the term "mindset" only in the title of the paper, which appears somehow awkward and leaves the definition of "mindset" unclear.
In order to improve this, we will use the term "attitudes" instead, which is way more clear and connects better to the article itself. We hope that this clears things up for the readers of the paper.